# Application of Microbial-Induced Calcium Carbonate Precipitation in Wave Erosion Protection of the Sandy Slope: An Experimental Study

**Yilong Li [1], Qiang Xu [2,\*], Yujie Li [2], Yuanbei Li [2] and Cong Liu [3]**

[1]  College of Civil Engineering and Architecture, Zhejiang University, Hangzhou 310058, China
[2]  Ocean Academy, Zhejiang University, Zhoushan 316021, China
[3]  Southwest Technology and Engineering Research Institute, Chongqing 400039, China
*   Correspondence: qiangxu@zju.edu.cn

**Abstract:** Sandy slope erosion leads to coast degradation and exacerbates coastal zone instability and failure. As an eco-friendly engineering technology, microbial-induced calcium carbonate precipitation (MICP) can provide a protection method against sandy slope erosion. In this study, a series of flume tests were conducted to investigate the wave erosion resistance of the MICP-treated sandy slope. The penetration tests were conducted to measure the slope surface strength, and the calcium carbonate content was evaluated by the acid washing method. The scanning electron microscope (SEM) was employed to study the microstructures of MICP-treated sand particles. In addition, the influence of MICP treatment on the wave shape and the excess pore water pressure was also analyzed. Results show that after four MICP treatments, the erosion resistance of the slope is significantly promoted, and no apparent erosion occurs after wave actions. The penetration resistance is also improved after MICP treatments, and the maximum penetration resistance of untreated and four-time MICP-treated slopes are about 0.14 MPa and 2.04 MPa, respectively. The calcium carbonate content on the slope surface can reach 7%. SEM analyses indicate that the intergranular bridging calcium carbonate crystals promote the wave erosion resistance of the sandy slope.

**Keywords:** microbial-induced calcium carbonate precipitation (MICP); sandy slope; erosion resistance; penetration resistance; calcium carbonate content; microstructures; wave shape; excess pore water pressure

## 1. Introduction

The coastal zone is one of the most active areas for human activities, and its sustainable development has great environmental and economic value. However, coastal vulnerability, i.e., the susceptibility of coastal zone being affected by erosion processes under the strong hydrodynamic action [1–3], has been a huge problem around the world. The strong hydrodynamic impaction tends to cause coastal zone degradation and the destruction of infrastructures. As an important part of the coastal zone, the sandy slope is particularly sensitive to external influences. Waves are one of the most obvious hydrodynamic conditions affecting the topography of the sandy slope [4,5]. Under wave actions, sand particles are transported and detached from the slope [6], leading to severe erosion of the sandy slope and coastal zone degradation [7,8]. Taking the potential disasters into account, effective protection techniques for the coastal sandy slope are urgently demanded to promote their erosion resistance.

At present, some structures, e.g., breakwaters, spur dikes, sea dikes, etc., are mainly adopted to mitigate sandy slope erosion. These man-made concrete structures can promote erosion resistance of the slope and have a long service life. However, the original balance of sediment transport around the slope is changed, which leads to new problems, e.g., the slope erosion may be accelerated in front of the seawall due to toe scouring. In addition, the environment can be polluted by the chemical substances in the concrete, and greenhouse

gases are emitted during cement production [9,10]. The chemical grouting methods have their limitations such as being rather expensive and limited injection distance. More notably, most chemicals are toxic and environmentally harmful [11]. The membrane structure is another choice to reduce slope erosion, but it has the disadvantages of high cost and disturbance to the infrastructures [12–15]. Therefore, it is necessary to find a sustainable coastal erosion protection technique, and the microbial approach may provide a solution to this problem.

By increasing the calcium carbonate content, microbial-induced calcium carbonate precipitation (MICP) is capable of increasing the strength and stiffness of granular soils [16–18]. The microorganisms secreting the enzyme urease are used in most studies on MICP, these microorganisms can supply urease and hydrolyze urea into ammonium and carbonate ions [19–23]. In the presence of calcium ions, generated carbonate ions precipitate as calcium carbonate crystals. Additionally, the precipitated calcium carbonate crystals accumulate between sand particles to form cementing bonds [24–27]. The reaction equations of the MICP process are as follows:

$$NH_2\text{-}CO\text{-}NH_2 + 2H_2O \rightarrow 2NH_4^+ + CO_3^{2-} \tag{1}$$

$$CO_3^{2-} + Ca^{2+} \rightarrow CaCO_3\downarrow \tag{2}$$

Compared with traditional coastal erosion protection techniques, MICP technology can be conducive to ecological protection and utilization improvement of land resources.

MICP technology has been applied to soil erosion control. Cheng et al. [28] found that the loess slope treated by MICP could resist the impact of raindrops as well as runoff erosion. Salifu et al. [29] tested the effectiveness of MICP on slope erosion mitigation, and the results indicated that negligible sediment erosion appeared on the slope as tidal processes occurred. Jiang et al. [30] reported that the MICP treatment could reduce the internal erosion and volumetric contraction of sand–clay mixtures. Wang et al. [31] found that the polymer-modified MICP treatment method could form a uniform soil crust in the surficial region and reduce the erodibility of sands. Moreover, MICP was also used for scour protection around the monopile by Li et al. [32], and the scour did not occur after four treatments.

Recently, the MICP technique has also been investigated to protect the sandy slope against erosion. Saitis et al. [33] and Daryono et al. [34] cemented the beach sediments into artificial beachrocks by the MICP method, and they found that the calcite crystals or aragonite crystals developed in the MICP process played a key role in creating the artificial beachrocks. The artificial beachrocks could share the same properties as the natural beachrock, and the MICP method could be an eco-friendly method to encounter beach erosion. Nayantara et al. [35] reported that the MICP technique can improve resistance to beach erosion without disturbing the hydraulics of soil samples. Kou et al. [36] demonstrated that the MICP percolation treatment method is a feasible approach to stabilizing coastal slopes, and it was found that slope erosion can be significantly reduced after 2–4 times MICP treatment. Shahin et al. [37] used a hydraulic model test to investigate the feasibility of MICP for preventing coastal erosion in shallow waters, and the results indicated that the MICP technique could control the coastal erosion of a 45° sandy slope to within 5%. Tsai et al. [38] indicated that the proper MICP treatment could mitigate sandy slope erosion under various wave conditions, and the MICP treatment could reduce sandy slope erosion up to 33.9% of the maximum scour depth. As for existing studies on the MICP protection of slope erosion [35–38], these model scales are too small to reflect the real cases, e.g., the boundary effect in small-scale tests is more obvious than that of the large-scale tests, and a small slope scale may lead to inaccurate sensor measurement in the tests, thus causing large test errors. In addition, the key parameters, e.g., the wave shape and the pore water pressure, have not been studied for the slopes with MICP treatments.

In this paper, a series of flume tests were conducted to investigate the wave erosion resistance of the MICP-treated sandy slope. Additionally, to further explore the effect of

MICP treatment, the penetration tests were conducted on the slope surface, and the calcium carbonate content was measured by the acid washing method. The microstructures of MICP-treated sand particles were analyzed by the scanning electron microscopy (SEM) method to reveal the mechanisms of MICP treatment. Moreover, the influence of MICP on the wave shape and the excess pore water pressure was also studied.

## 2. Materials and Methods

### 2.1. Materials

The Fujian Sand, mainly composed of quartz [39,40], was adopted in this study. The particle size distribution of the Fujian Sand is shown in Figure 1a, this sand has a specific gravity ($G_s$) of 2.633, a mean grain size ($d_{50}$) of 0.17 mm, a uniformity coefficient ($C_u = d_{60}/d_{10}$) of 1.57, and a curvature coefficient [$C_c = (d_{30})^2/(d_{60} \times d_{10})$] of 0.96. The minimum and maximum void ratios of the Fujian Sand are 0.607 and 0.952, respectively. The typical SEM image of the Fujian Sand is shown in Figure 1b, the particle shape is regular, and the surface is relatively smooth.

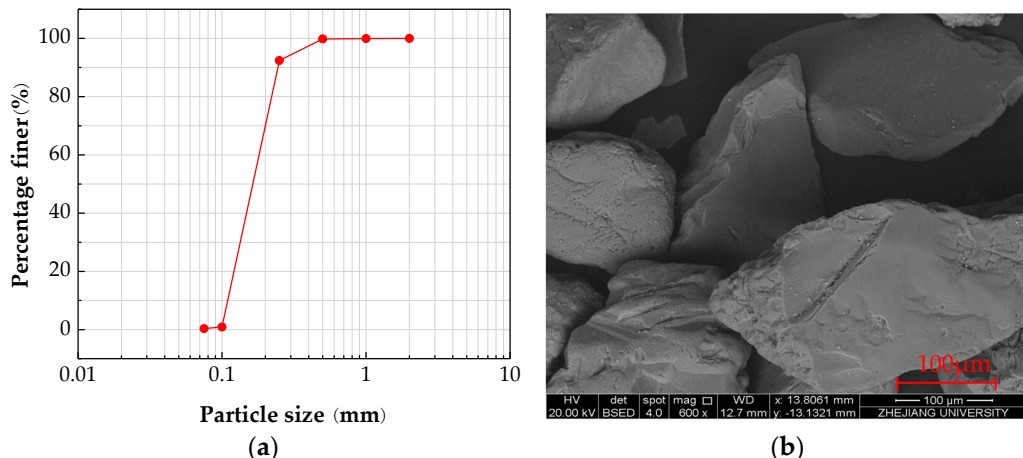

**Figure 1.** (**a**) Particle size distribution curve; and (**b**) SEM image of the Fujian Sand.

### 2.2. Experiment Set-Up

A series of model tests were carried out in a flume with a length of 69 m, a width of 1.2 m, and a height of 1.6 m (Figure 2a). The wall of the flume was made of transparent tempered glass. As shown in Figure 2b, a push plate wave generator was used for wave generation, which had a regular period of 0.5~5.0 s to make the wave. When the water depth was 1 m, the maximum regular wave height could be 0.4 m. As illustrated in Figure 2c, a sandy slope with a length of 4 m and a height of 0.6 m was located in the flume, and the tail of the sandy slope was supported by a brick wall. A rigid shunt plate with a length of 6 m and a height of 1.2 m was arranged on the central axis of the flume. The relative compactness of the sand was controlled to be 65%. Considering the uniformity of the slope, the sand was put into the water tank, and then it was put into the flume after being uniformly stirred with water. After the MICP treatment, the slope was soaked in water for 48 h to ensure the sand was fully saturated, then the wave was made to start the flume test. The layout of wave gauges is shown in Figure 2c. 2#, 3# wave gauges were used to monitor the wave shape in the area with the most severe wave actions (wave surf zone).

To monitor the pore water pressure within the slope under wave actions, pore pressure sensors were installed around the wave surf zone. As depicted in Figure 3, the first column of pore pressure sensors was set at a distance of 2.4 m from the slope toe. The configuration of 4 columns with a 0.2 m distance was adopted to arrange the pore pressure sensors. There were 12 pore pressure measurement points in total, and the pore pressure sensors were numbered in sequence. The first row of pore pressure sensors was placed on the slope surface.

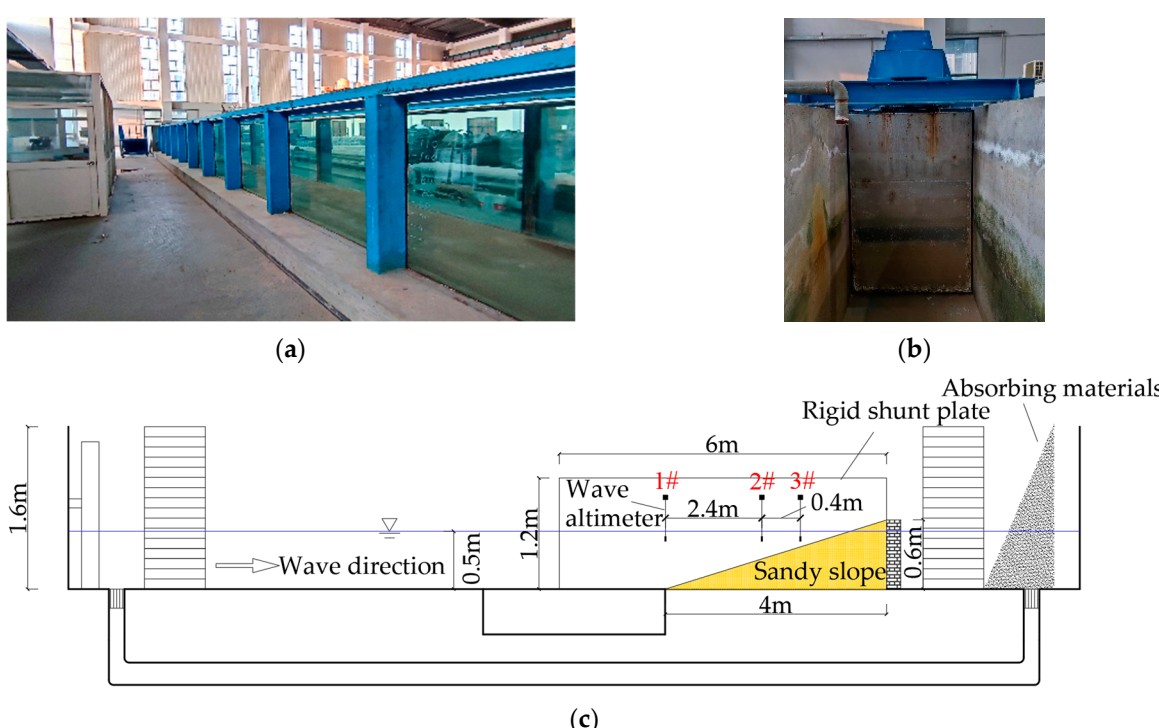

**Figure 2.** Large section flume and experiment set-up: (**a**) large section flume; (**b**) push plate wave generator; (**c**) the experiment set-up.

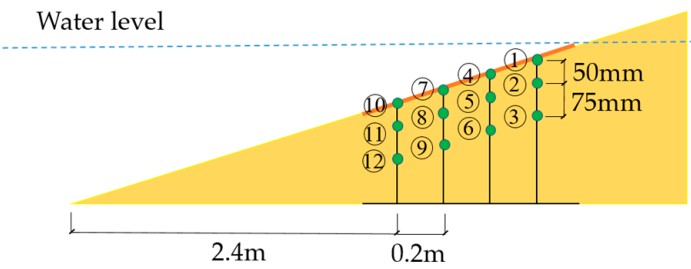

**Figure 3.** Schematic diagram of pore pressure sensors layout.

The test conditions are summarized in Table 1. To investigate the effect of MICP treatment times on slope erosion protection, the times of MICP treatment were set to 0, 2, and 4 under the wave height of 10 cm. To explore the effect of MICP on slope erosion protection under different wave heights, wave heights of 8 cm and 12 cm were adopted in the flume tests. The wave erosion duration of 60 min was generated under all test conditions.

**Table 1.** Conditions for flume tests.

| Test Group | Wave Height (cm) | Wave Period (s) | MICP Treatment Times |
|---|---|---|---|
| 1 | 10 | 1 | 0 |
| 2 | 10 | 1 | 2 |
| 3 | 10 | 1 | 4 |
| 4 | 8 | 1 | 0 |
| 5 | 8 | 1 | 4 |
| 6 | 12 | 1 | 0 |
| 7 | 12 | 1 | 4 |

### 2.3. MICP Treatment Method

The *Bacillus pasteurii*, a urease-producing bacterium, was used in the test. In this study, the bacteria were cultivated in an ATCC-specified medium with a 2% vaccination ratio, this medium consisted of 20 g/L yeast extract, 10 g/L $NH_4Cl$, and 0.13 M Tris buffer with a pH value of 9.25 [41,42]. In order to avoid the contamination in the process of bacterial culture, the prepared culture medium should be placed in an autoclave, sterilized for 20 min at 121 °C, and then cooled for later use. As shown in Figure 4, the vaccinated culture medium was grown in a shaking incubator operated at 30°C and 180 rpm for 24–28 h until the final optical density (OD600) could reach approximately 1.7–2.0 and the enzyme activity at 20.3–25.2 U/mL (1 U = 1 µmol urea hydrolyzed/min).

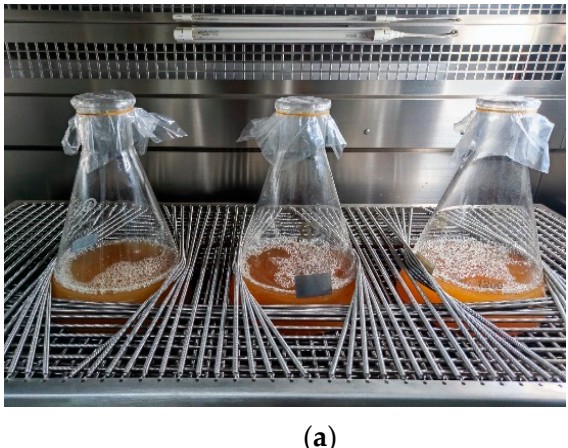
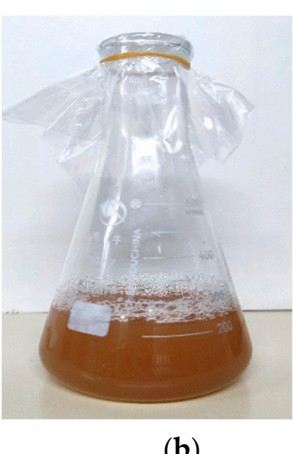

(**a**)　　　　　　　　　　　　　　　　　　(**b**)

**Figure 4.** (**a**) Bacterial suspension cultured in shaking incubator; (**b**) culture medium after completion of culture.

As illustrated in Table 2, to improve the utilization rate of calcium ions, the concentration of $CaCl_2$ and urea in the cementation solution was set to be 0.5 M and 1 M, respectively. In addition, 10 g/L tryptone and 3 g/L beef extract were added to the cementation solution to provide nutrients for bacteria. MICP treatment was performed on the area with the most severe wave actions (wave surf zone) (Figure 5). The two-phase spraying method [43] was employed in this investigation for MICP treatment. Firstly, to reduce the impact of flow during spraying, a layer of geotextile was laid on the slope surface. Then, the bacterial suspension was evenly sprayed with 1 mL/cm$^2$ at a flow rate of 20 mL/s, and it was left for 6 h to allow the bacteria fully adsorbed onto the sand particles. After that, the cementation solution was sprayed with 2 mL/cm$^2$ at a flow rate of 20 mL/s, and then the geotextile was removed. After standing for 24 h, the MICP treatment was completed. The bacterial suspension and the cementation solution percolated freely under its gravity and could flow from the top of the treated area to the toe.

**Table 2.** Composition of cementation solution (g/L).

| Substance | Content (g/L) |
|:---:|:---:|
| $CaCl_2$ | 55 |
| Urea | 60 |
| Tryptone | 10 |
| Beef extract | 3 |

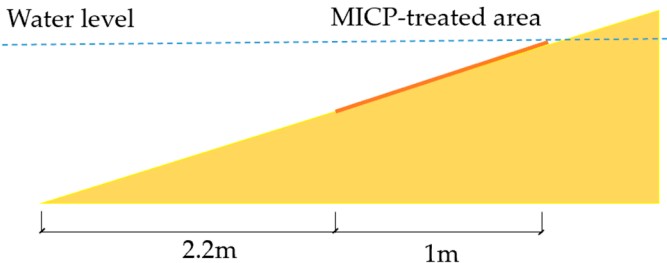

**Figure 5.** MICP-treated area of the sandy slope.

*2.4. Penetration Resistance Test*

To quantitatively evaluate the strength of the MICP-treated sandy slope, the penetration resistance tests were carried out on the MICP-treated area by an electronic pocket penetrometer (MPT with a measuring range of up to 2660 kPa) before wave impaction [44]. The penetration position was selected to be uniformly distributed along the slope surface. As depicted in Figure 6a, 16 penetration tests were conducted on the slopes with and without MICP treatment. The distance between adjacent penetration points was determined to be 12 cm, being more than 12 times the diameter of the penetration cone to minimize interference. The 30-degree cone tip was vertically penetrated into the sand to a depth of 10 mm (Figure 6b).

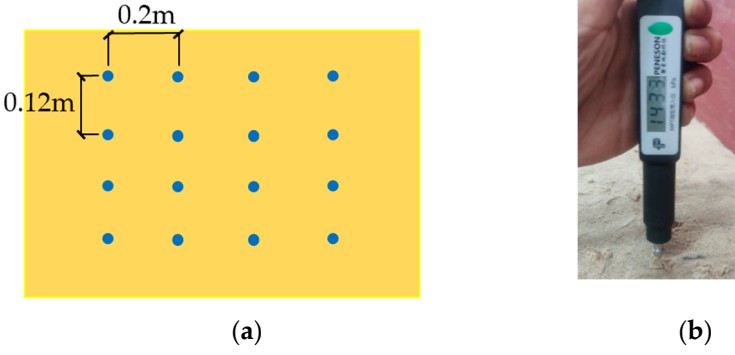

(**a**)                                                                                        (**b**)

**Figure 6.** Penetration resistance tests: (**a**) position diagram of penetration points (top view); (**b**) penetration resistance test method.

*2.5. Calcium Carbonate Content Test*

The acid washing method was used to determine the content of the produced calcium carbonate [45]. After wave actions, the MICP-treated sand was sampled. It was illustrated in Figure 7 that 8 cross sections evenly distributed along the MICP-treated area were sampled, and 4 samples were carefully extracted at different depths for each cross-section. The samples were dried in an oven to reach a constant weight, and then 1 M hydrochloric acid (HCl) was added to the samples until no bubbles were generated. After being washed with distilled water, the samples were put into the oven for 24 h at 110 °C. The calcium carbonate content ($C_i$) can be expressed as:

$$C_i = (m_{1i} - m_{2i})/m_{2i} \times 100\% \tag{3}$$

where $C_i$ is the calcium carbonate content, $m_{1i}$ is the mass of sample $i$ before acid washing, and $m_{2i}$ is the mass of sample $i$ after acid washing. For a cross-section, the calcium carbonate content of each depth was determined by the average calcium carbonate content of 4 samples at that depth.

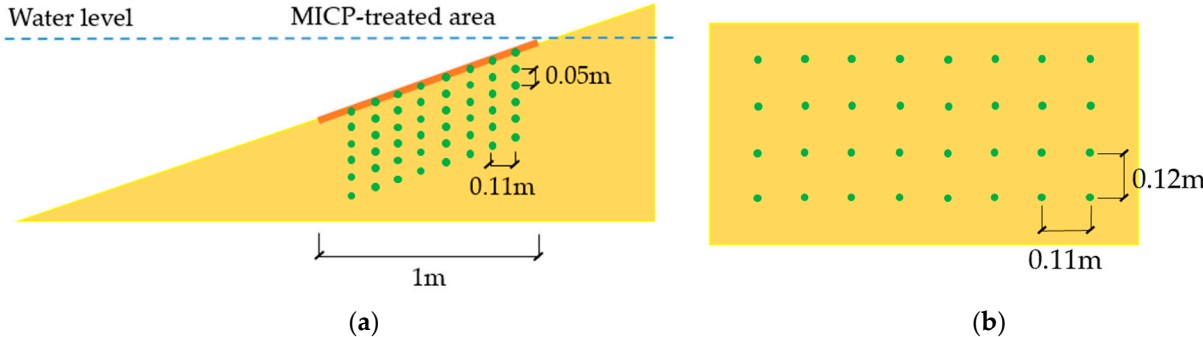

**Figure 7.** Schematic diagram of measurement points for calcium carbonate content: (**a**) side view; (**b**) top view.

### 2.6. SEM Analyses

After wave impaction, SEM analyses were carried out by a field emission scanning electron microscope (FEI QUANTA FEG 650), which was equipped with a secondary electron probe, a backscattered electron probe, and an X-ray energy spectrum probe. Additionally, the microstructures, phase compositions, and elemental compositions of materials could be observed and analyzed. Before SEM analyses, the samples must be sputter-coated with a layer of gold by a vacuum ion sputtering instrument to ensure uniform surface conductivity and avoid the electric phenomenon.

## 3. Results and Discussion

### 3.1. Erosion Characteristics of the MICP-Treated Area

When the wave propagates on the slope, the decrease in water depth shortens the wavelength and increases the wave height, and the speed of water particles' movement is greater than that of wave propagation. Upon reaching a physical limit, the wave breaks and will not maintain its shape. The wave crest rolls forward and falls, impacting the water flow on the slope and splashing a lot of water. For the slope without MICP treatment, the violent movement of the water rolls up the sediment on the slope. In the process of backflow, the flow still has a large sand-carrying capacity, taking away the sediment carried during climbing. Additionally, the shear stress of water flow causes a large number of sand particles to be suspended. The flow speed gradually decreases, and the sand-bearing flow dissipates a lot of energy in the near-shore hydraulic jump area and causes sediment deposition. Therefore, the scour pit is formed near the slope shoreline, and the sand bar is formed on the offshore side of the scour pit due to a large amount of sediment deposition.

When the wave height is 10 cm, the erosion characteristics of sandy slopes with different MICP treatment times are depicted in Figure 8. Without the MICP treatment, the slope is severely eroded, and the obvious scour pit appears (Figure 8a). The scour pit depth is determined as the maximum erosion depth of the slope on the side of the shunt plate, which is 6 cm. After two MICP treatments, a small, cemented block is formed in the erosion area, and the depth of the scour pit is reduced to 5 cm. However, the erosion is still obvious, and the protective effect is inefficient (Figure 8b). However, after four MICP treatments, a solid protective layer is formed on the slope surface. The MICP-treated area can still maintain great integrity after 60 min of wave action, and no apparent erosion occurs (Figure 8c).

When the wave heights are 8 cm and 12 cm, serious erosion occurs on the slope without MICP treatment, the depth of the scour pits in the wave surf zone is 5 cm and 7 cm, respectively (Figure 9a,c). After four MICP treatments, there is basically no erosion in the MICP-treated area of the slope (Figure 9b,d). In summary, after four MICP treatments, the slope has a strong anti-wave erosion ability.

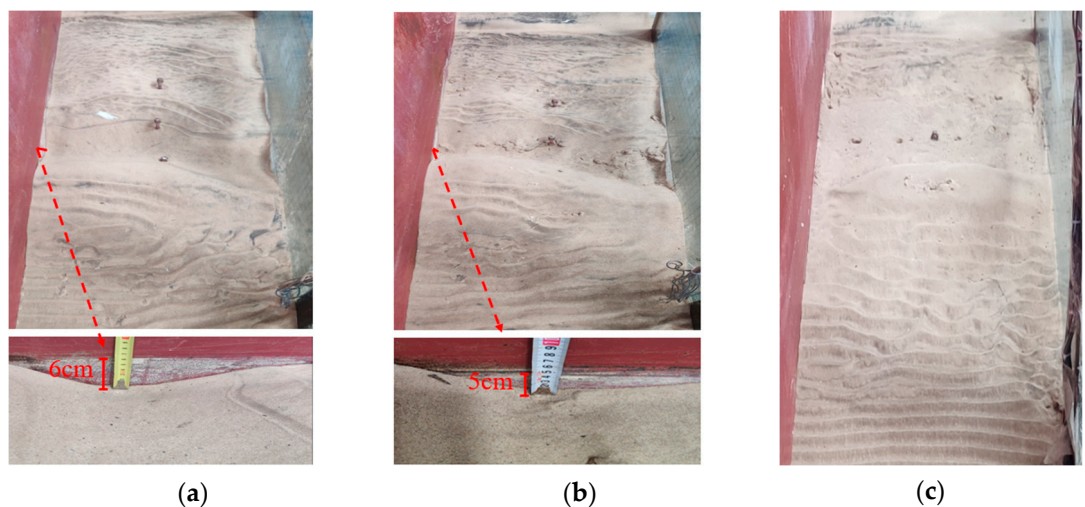

**Figure 8.** Erosion characteristics of the slope: (**a**) without MICP treatment; (**b**) 2-time MICP treatments; (**c**) 4-time MICP treatments (wave height is 10 cm).

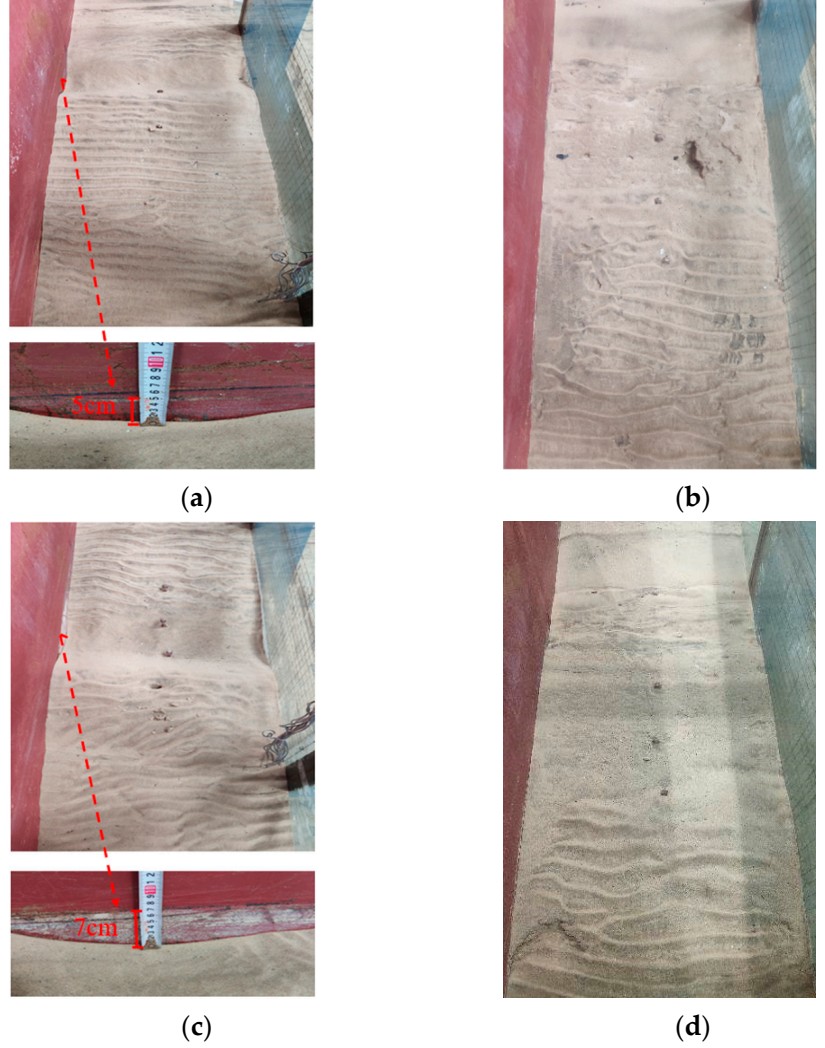

**Figure 9.** Erosion characteristics of the slope: when the wave height is 8 cm (**a**) without MICP treatment, (**b**) 4-time MICP treatments; when the wave height is 12 cm (**c**) without MICP treatment, (**d**) 4-time MICP treatments.

The difference between two and four MICP treatments can be explained as follows. In the first two MICP treatments, the permeability is not significantly decreased in the shallow layer of the slope. Large amounts of bacterial suspension and cementation solution permeate into the bottom of the slope from the surface. Therefore, a small amount of bacterial suspension and cementation solution remains on the surface, resulting in a poor reinforcement effect on the slope surface. However, with the increase in treatment times, e.g., four times, the permeability of the shallow layer of the slope is significantly decreased, and more bacterial suspension and cementation solution remain on the slope surface. In addition, the *Bacillus pasteurii* is an oxyphilic bacterium, and bacteria on the surface layer of the slope are more likely to be exposed to oxygen. Therefore, with the increase in treatment times, the improvement of the reinforcement effect of the slope surface is more obvious than that of deeper depth.

### 3.2. Penetration Resistance and Calcium Carbonate Content

#### 3.2.1. Penetration Resistance

The penetration resistance test results are depicted in Figure 10. After the MICP treatment, the strength of the slope surface is improved. In particular, after four MICP treatments, the maximum penetration resistance of the slope surface can reach up to 2.04 MPa, and the maximum penetration resistance of the slope without MICP treatment is just 0.14 MPa. However, after two MICP treatments, the strength of the slope surface is not significantly improved, and the maximum penetration resistance is about 0.51 MPa. In the process of MICP treatment, a large amount of bacterial suspension and cementation solution flows to the toe of the MICP-treated area due to gravity. Especially after multiple MICP treatments, the permeability of the slope surface is decreased, so that the subsequent bacterial suspension and the cementation solution are easier to flow down the slope. In essence, there is more bacterial suspension and cementation solution at the toe of the MICP-treated area, causing the penetration resistance to decrease from the toe to the top of the MICP-treated area. It also can be obtained from Figure 10 that with the increase in treatment times, the dispersion of the penetration resistance on each cross-section is increased, which may be caused by the uneven flow of bacterial suspension and cementation solution under the action of gravity. The content of bacterial suspension and cementation solution of the same cross-section is different, leading to inconsistent reinforcement effects and increased strength dispersion.

It can be concluded from penetration resistance tests that after two MICP treatments, the strength of the slope surface is not significantly enhanced, so the erosion protection effect is weak. However, after four MICP treatments, the slope surface strength is significantly improved, so the slope anti-wave erosion ability is markedly promoted.

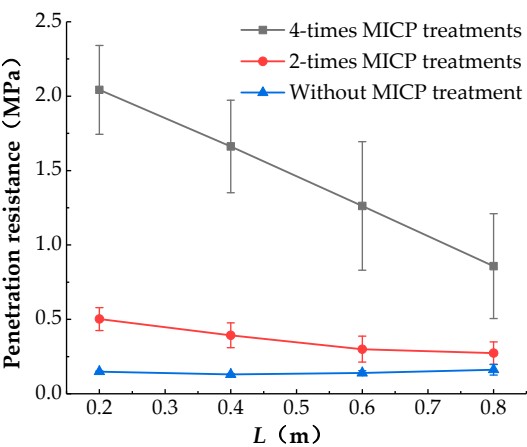

**Figure 10.** Penetration resistance of MICP treated area.

### 3.2.2. Calcium Carbonate Content

The test results of calcium carbonate content are summarized in Figure 11, where *L* is the horizontal distance between the measurement points and the toe of the MICP-treated area. As mentioned above, in the process of MICP treatment, the bacterial suspension and the cementation solution flow from the top of the MICP-treated area to the toe due to gravity. Therefore, the calcium carbonate content of the MICP-treated area decreases from the toe to the top of the slope. In addition, it can be seen from Figure 11 that the calcium carbonate content decreases with depth, and it decreases by a larger gradient in the shallow layer of the slope. With the increase in MICP treatment times, the permeability of the slope is gradually decreased, resulting in difficult infiltration of bacterial suspension, cementation solution, and oxygen. Therefore, the difference in bacterial suspension, cementation solution, and oxygen content between the deep and the shallow layer of the slope is increased, leading to the increased difference of calcium carbonate content. In addition, due to the greater reduction in the permeability in the shallow layer of the slope, the reduction gradient of calcium carbonate in the shallow layer of the slope is larger.

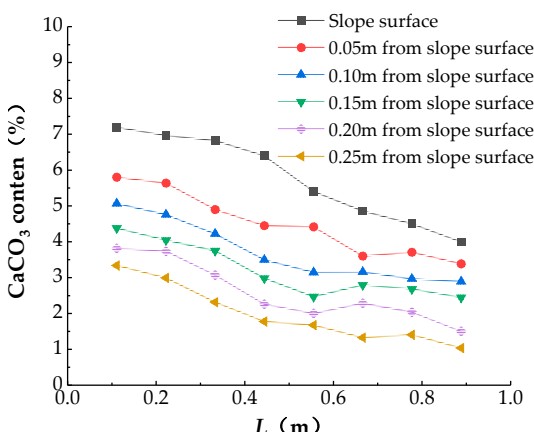

**Figure 11.** Calcium carbonate content of the MICP treated area (4-time MICP treatments).

It can be observed from Figure 12 that after four MICP treatments, the sand body is reinforced into a unified whole with great integrity. In the first two MICP treatment times, a large amount of bacterial suspension and cementation solution permeates into the deep layer of the sand body. Consequently, the deep layer of the slope is effectively reinforced in the first two MICP treatment times.

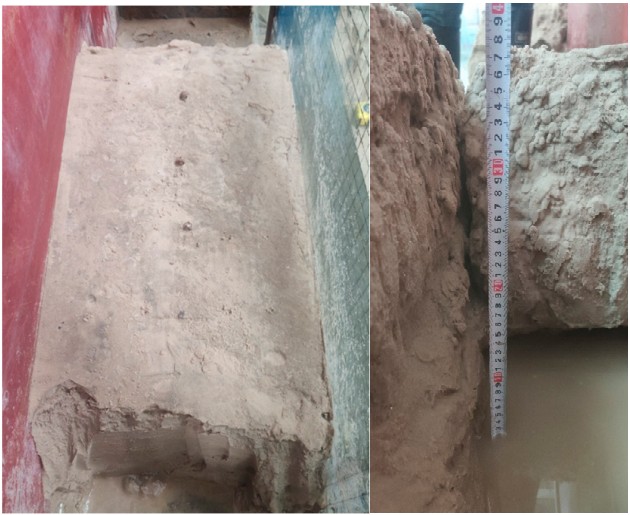

**Figure 12.** MICP-treated sand body (4-time MICP treatments).

In conclusion, the reinforcement effect of the deep layer of the slope is improved more obviously in the previous MICP treatments, and with the increase in MICP treatment times, the reinforcement effect of the slope surface is improved more obviously.

### 3.3. Microstructures of MICP-treated Sand Particles

Figure 13 shows the SEM images of sand particles after four MICP treatments, and it is obvious that lots of calcium carbonate crystals are formed on the surface of the sand particles. Furthermore, it can be clearly observed from the SEM images that calcium carbonate crystals play three main roles in the cementation: (1) intergranular bridging to enhance the strength of the sand body; (2) particle wrapping to increase the particle size and the surface roughness; (3) pore filling to reduce the permeability of sand body. Especially, the intergranular bridging calcium carbonate crystals improve the ability of the slope to resist wave erosion.

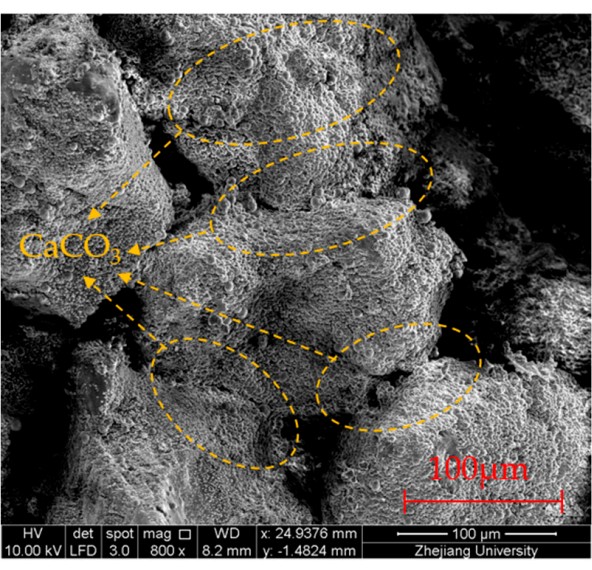
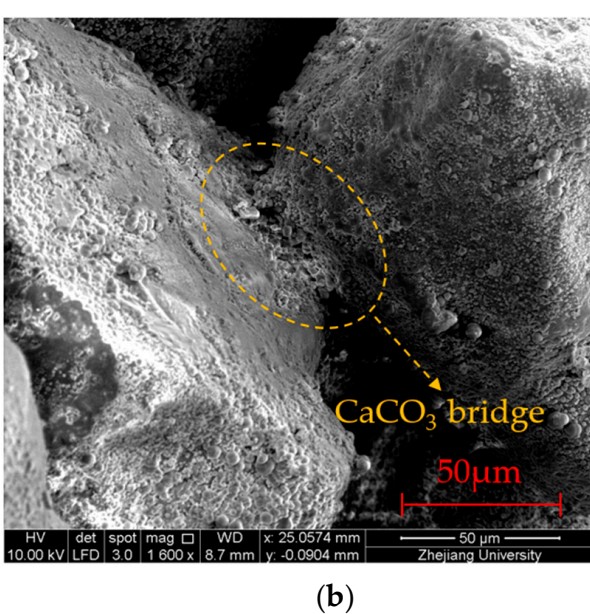

(**a**)  (**b**)

**Figure 13.** SEM images of MICP-treated sand particles (4-time MICP treatments): (**a**) image of sand particles magnified 800 times; (**b**) image of sand particles magnified 1600 times.

### 3.4. Wave Shape and Excess Pore Water Pressure

#### 3.4.1. Wave Shape

The wave shape can be influenced by topographic changes on the slope during erosion. For slopes without the MICP treatment, the water depth at the sand bar decreases. When the wave propagates to the sand bar, the wave shape becomes unstable, causing the wave breaking point to be moved forward. However, for slopes with four MICP treatments, no apparent erosion occurs in the MICP-treated area, so the position of the wave breaking point is basically not changed.

Figure 14 shows the wave shape at each wave gauge on the slopes with and without MICP treatment. It can be found that, under the wave height of 8 cm and 10 cm, the wave shape on slopes with four MICP treatments are more regular than that on untreated slopes. When the wave height is 12 cm, the wave is more likely to break in the shallow water. Therefore, the wave has already broken before propagating to the 3# wave gauge, regardless of whether the slope is treated by MICP or not.

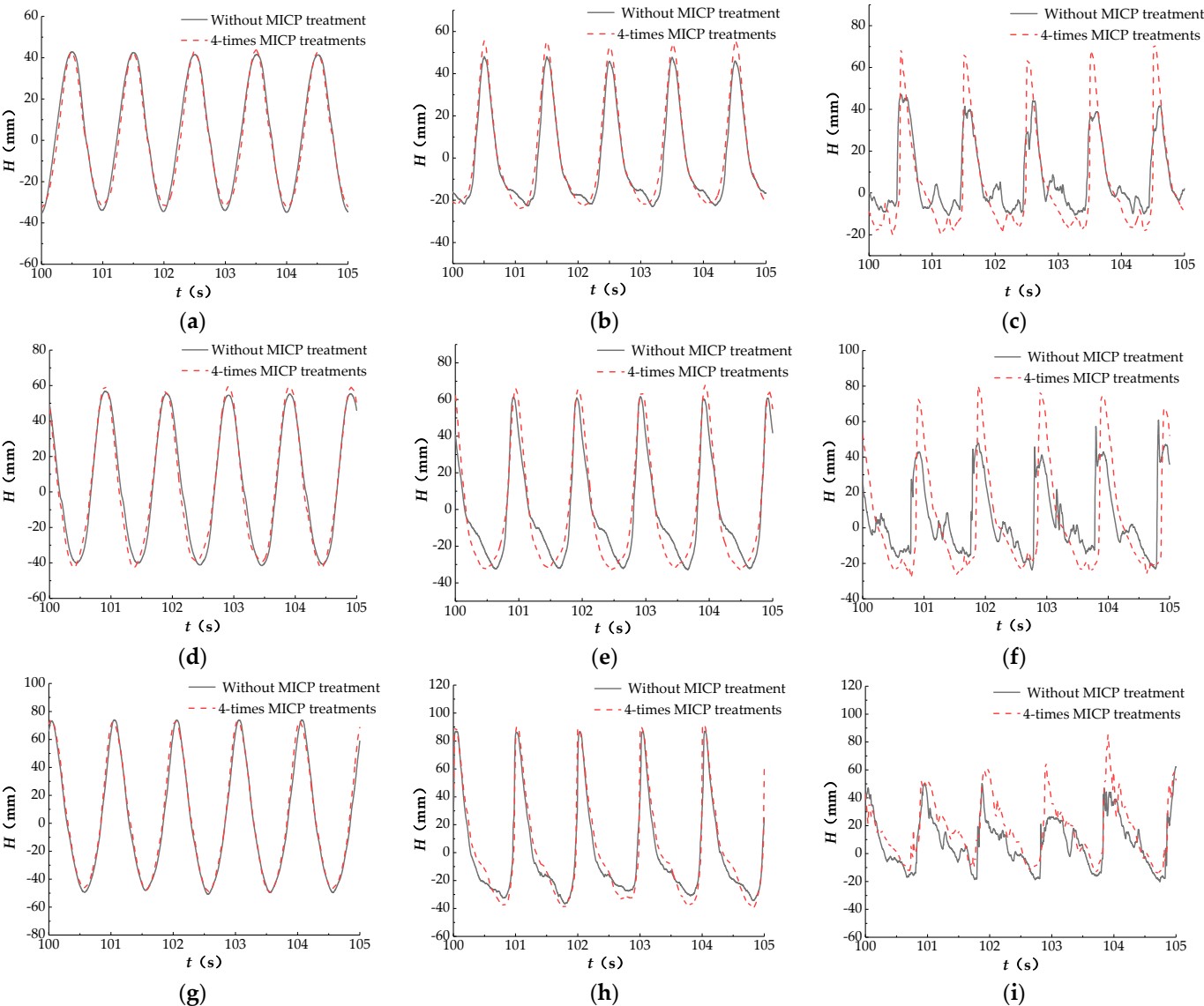

**Figure 14.** Wave shape at each wave gauge position on slopes with and without MICP treatment: when the wave height is 8 cm (**a**) 1#, (**b**) 2#, (**c**) 3#; when the wave height is 10 cm (**d**) 1#, (**e**) 2#, (**f**) 3#; when the wave height is 12 cm (**g**) 1#, (**h**) 2#, (**i**) 3#.

### 3.4.2. Excess Pore Water Pressure

The normalized excess pore water pressure amplitude at each measurement point is shown in Figure 15, where $z$ represents the depth of the measurement point, $p/p_0$ represents the ratio of the excess pore water pressure amplitude at the measurement point to the wave pressure amplitude on the slope surface. It can be obtained from Figure 15 that with the increase in the depth, the excess pore water pressure in the slope decreases. In addition, after four MICP treatments, the accumulation of calcium carbonate crystals between sand particles reduces the permeability of the sand body, which reduces the ability to transfer water pressure inside the slope. Therefore, it is also found from Figure 15 that, after four MICP treatments, the excess pore water pressure gradient at the shallow layer of the slope is significantly increased compared with the slope without MICP treatment.

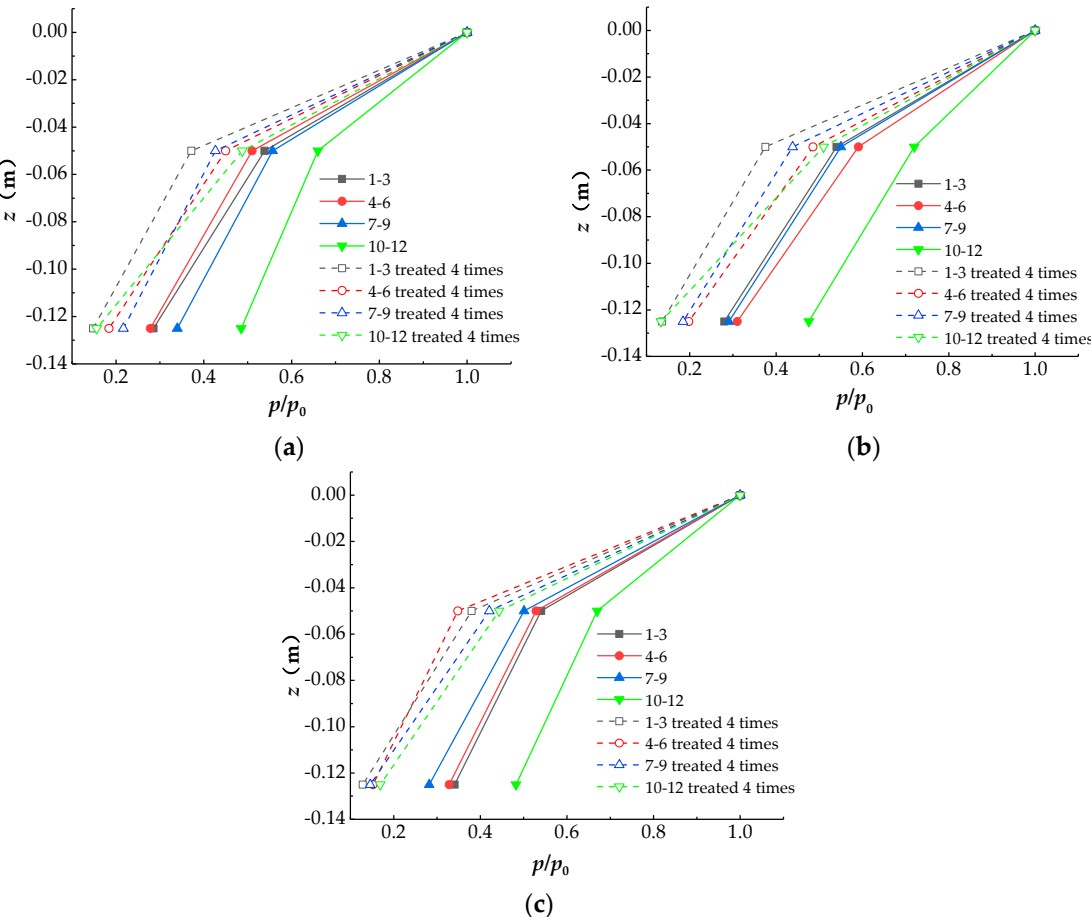

**Figure 15.** The normalized excess pore water pressure amplitude at measurement points: (**a**) wave height is 8 cm; (**b**) wave height is 10 cm; (**c**) wave height is 12 cm.

## 4. Conclusions

In this paper, a series of flume tests were conducted to obtain the wave erosion resistance of the MICP-treated sandy slope. The penetration resistance and calcium carbonate content of the MICP-treated area were measured, and the SEM analyses were conducted to investigate the microstructures of MICP-treated sand particles. In addition, the influence of MICP treatment on the wave shape and the excess pore water pressure was also evaluated. The main conclusions can be derived:

(1) After two MICP treatments, the erosion resistance of the slope is not significantly improved, because the majority of bacterial suspension and cementation solution permeates into the bottom of the slope. However, with the increase in treatment times, the permeability of the slope decreases. Thus, more bacterial suspension and cementation solution remain in the shallow layer of the slope, and the erosion resistance of the slope is improved rapidly. After four MICP treatments, the topography of the MICP-treated area (wave surf zone) basically remains unchanged under wave actions.

(2) The strength of the slope surface is slightly increased after two MICP treatments. However, after four MICP treatments, the strength of the slope surface is significantly promoted, and the penetration resistance is up to 2.04 MPa. After four MICP treatments, the calcium carbonate content of the slope surface can reach up to 7%, and the calcium carbonate content decreases with depth due to the uneven reaction.

(3) The microstructures of MICP-treated sand particles indicate that the intergranular bridging, particle wrapping, and pore-filling calcium carbonate crystals are produced by MICP. Especially, the intergranular bridging calcium carbonate crystals enhance the strength of the sand body and improve the ability of the slope to resist wave erosion.

(4) For the slope without the MICP treatment, the wave breaking point is moved forward during wave erosion, as the sand bar is formed under wave actions. However, after four MICP treatments, the topography changes of the wave surf zone are unapparent, so the position of the wave breaking point is basically not changed. In addition, after MICP treatments, the excess pore water pressure gradient at the shallow layer of the slope is increased due to the decreased permeability.

(5) This study was conducted indoors and cannot truly reflect the actual situation of the project site. It is recommended that MICP reinforcement tests on sandy slopes should be conducted at the coastal site to explore the wave erosion resistance of MICP-treated sandy slopes. In addition, in the harsh environment of the coastal zone, it is necessary to focus on improving the durability of MICP-treated sand bodies.

**Author Contributions:** Conceptualization, Y.L. (Yilong Li) and Q.X.; methodology, Y.L. (Yilong Li); investigation, Y.L. (Yujie Li); resources, Q.X.; writing—original draft preparation, Y.L. (Yuanbei Li); writing—review and editing, Q.X.; funding acquisition, Q.X. and C.L. All authors have read and agreed to the published version of the manuscript.

**Funding:** This work was financed by the Southwest Institute of Technology and Engineering Cooperation fund (HDHDW5901010101). The authors are grateful for the receipt of this fund.

**Institutional Review Board Statement:** Not applicable.

**Informed Consent Statement:** Not applicable.

**Data Availability Statement:** All relevant data are presented in the manuscript.

**Conflicts of Interest:** The authors declare no conflict of interest.

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
