# Peer review of "Application of Microbial-Induced Calcium Carbonate Precipitation in Wave Erosion Protection of the Sandy Slope: An Experimental Study"

_sustainability, doi:10.3390/su142012965_

Round 1

Reviewer 1 Report

General comments:

The authors have presented an article on the application of MICP in the protection of sandy slope. This article belongs to the field of Sustainability and presents unprecedented research results. 

Specific comments:

1.           It is recommended to strengthen the MICP treatment method description (section 2.3). For example, the water level during the spraying process, the geotextile removed time, and the spraying rate. The above might be important information for developing the MICP application for beach protection.

2.           Figure 7, 8 (in Line 187) seems to have a wrong figure number; it is recommended to adjust it.

3.           There is a lack of comprehensive comparison in Section 3.1. For example, it is not easy to distinguish the erosion difference between each case from the picture. Suggesting a side view comparison of original and erosion profiles might help readers to understand.

4.           It should be described clearly whether the results in sections 3.2 and 3.3 are after wave impaction or not.

5.           The wave profile in Section 3.4.1. recommends a more logical arrangement. For example, the same cases are arranged in the same column or row, which is convenient for readers to understand. Moreover, the images of waves acting on the sandy slope could be more helpful for the explanation.

Author Response

Response to Reviewer 1 Comments

General comments: The authors have presented an article on the application of MICP in the protection of sandy slope. This article belongs to the field of Sustainability and presents unprecedented research results.

Specific comments:

Point 1: It is recommended to strengthen the MICP treatment method description (section 2.3). For example, the water level during the spraying process, the geotextile removed time, and the spraying rate. The above might be important information for developing the MICP application for beach protection.

Response 1: Thank you very much for your comment.

The slope was treated by MICP in an unsaturated state, when there was no water in the flume. During the MICP treatment, the spraying rate of bacterial suspension and cementation solution was controlled to be about 20 mL/s. When the spraying of the cementation solution was finished, the geotextile was removed. The corresponding adjustments have been made in this paper.

Point 2: Figure 7, 8 (in Line 187) seems to have a wrong figure number; it is recommended to adjust it.

Response 2: Thanks for your valuable comments, the meaning here is unclear, and the author has redescribed this as “It was illustrated in Figure 7 that 8 cross sections evenly distributed along the MICP-treated area were sampled, ……”.

Point 3: There is a lack of comprehensive comparison in Section 3.1. For example, it is not easy to distinguish the erosion difference between each case from the picture. Suggesting a side view comparison of original and erosion profiles might help readers to understand.

Response 3: The slope surface morphology (especially the slope surface integrity and maximum scour depth) after wave erosion under different MICP-treatment times was compared to illustrate the erosion protection effect of MICP on the slope under wave actions. Due to the large size of the slope and the poor transparency of the flume glass, it is difficult to take a more accurate and clear side view of the slope. The full view of the erosion can be clearly captured by the pictures provided in this paper. And it is clearly illustrated by the pictures provided in this paper that after 4-times MICP treatment, the slope surface is basically not eroded under wave actions, and the flatness and integrity of the slope surface are good, while for the slopes with 0 or 2-times MICP treatments, the surface is seriously eroded. And the scour pit depth is determined as the maximum erosion depth of the slope on the side of the shunt plate, the maximum scour pit depth of the slopes with 4-times MICP treatments is negligible, while the slopes with 0 or 2-times MICP treatments have larger scour depth. Therefore, the pictures in this paper can clearly show that the slope has strong anti-erosion performance after 4-times MICP treatments.

Point 4: It should be described clearly whether the results in sections 3.2 and 3.3 are after wave impaction or not.

Response 4: The penetration resistance of the slope surface in 3.2.1 is measured before wave impaction, the calcium carbonate content in the slope in 3.2.2, SEM images of sand particles in 3.3 are measured after wave impaction[1]. The corresponding adjustments have been made in this paper.

References

  1. Kou, H.L.; Wu, C.Z.; Ni, P.P.; Jang, B.A. Assessment of erosion resistance of biocemented sandy slope subjected to wave actions. Ocean Res. 2020, 105, 102401.

Point 5: The wave profile in Section 3.4.1. recommends a more logical arrangement. For example, the same cases are arranged in the same column or row, which is convenient for readers to understand. Moreover, the images of waves acting on the sandy slope could be more helpful for the explanation.

Response 5: We have made corresponding adjustments in the text, putting the same cases in the same row to make it easier for readers to understand. Due to the poor transparency of the flume wall, it is impossible to clearly and accurately take pictures of waves acting on the sandy slope. Therefore, wave gauges were used to capture the wave shape, and the wave shape was compared under different test conditions. The conclusion can also be drawn more intuitively.

Reviewer 2 Report

Overall well written. The following comments should consider; 

-Please add a table to 2.3. MICP Treatment Method about the parameters considered during the tests.

Line 151, check the suffix. Better to check prefix and suffix throughout the whole manuscript.

Drawbacks, future recommendations can be added to conclution sections.

Author Response

Response to Reviewer 2 Comments

Point 1: Please add a table to 2.3. MICP Treatment Method about the parameters considered during the tests.

Response 1: Thank you very much for your valuable advice. To express more clearly, we added Table 2 which is “Composition of cementation solution.” to 2.3.

Table 2. Composition of cementation solution (g/L).

Substance

Content (g/L)

CaCl2

55

urea

60

tryptone

10

beef extract

3

Point 2: Line 151, check the suffix. Better to check prefix and suffix throughout the whole manuscript.

Response 2: Thank you very much for your valuable advice. We have checked prefix and suffix throughout the whole manuscript seriously.

Point 3: Drawbacks, future recommendations can be added to conclution sections.

Response 3: Thanks for your valuable advice. We added future recommendations in conclusion sections.

The future recommendations are as follows:

(5) This study was conducted indoors and cannot truly reflect the actual situation of the project site. It is recommended that MICP reinforcement tests on sandy slopes should be conducted at the coastal site to explore wave erosion resistance of MICP-treated sandy slopes. In addition, in the harsh environment of the coastal zone, it is necessary to focus on improving the durability of MICP-treated sand bodies.

Reviewer 3 Report

I have carefully read the paper and I believe that it and, believe it is a very promising work worth publishing. This study is conducted within hypothetical experimental contexts in respect of the application of MICP method using ureolytic bacteria for stabilization of sandy slopes against wave action. A very strong aspect of this study is the variety of analytical methods that were used. However, there are some issues with the language through the text suggesting revision.

I suggest the authors should include studies using microscopical and SEM-EDS analysis on MICP treated sand samples in order to improve the introduction.

suggested bibliography:

1) Saitis, G.; Karkani, A.; Koutsopoulou, E.; Tsanakas, K.; Kawasaki, S.; Evelpidou, N. Beachrock Formation Mechanism Using Multiproxy Experimental Data from Natural and Artificial Beachrocks: Insights for a Potential Soft Engineering Method. J. Mar. Sci. Eng. 2022, 10, 87. https://doi.org/10.3390/jmse10010087

2) Daryono, L.R.; Titisari, A.D.; Warmada, I.W.; Kawasaki, S. Comparative characteristics of cement materials in natural and artificial beachrocks using a petrographic method. Bull. Eng. Geol. Environ. 2019, 78, 3943–3958.

For specific comments, please refer in the manuscript.

Overall, my suggestion is that the paper should be accepted after major revisions.

Author Response

Response to Reviewer 3 Comments

Point 1: I suggest the authors should include studies using microscopical and SEM-EDS analysis on MICP treated sand samples in order to improve the introduction.

Suggested bibliography:

1) Saitis, G.; Karkani, A.; Koutsopoulou, E.; Tsanakas, K.; Kawasaki, S.; Evelpidou, N. Beachrock Formation Mechanism Using Multiproxy Experimental Data from Natural and Artificial Beachrocks: Insights for a Potential Soft Engineering Method. J. Mar. Sci. Eng. 2022, 10, 87. https://doi.org/10.3390/jmse10010087

2) Daryono, L.R.; Titisari, A.D.; Warmada, I.W.; Kawasaki, S. Comparative characteristics of cement materials in natural and artificial beachrocks using a petrographic method. Bull. Eng. Geol. Environ. 2019, 78, 3943–3958.

For specific comments, please refer in the manuscript.

Response 1: Thank you so much for providing two such valuable articles that have benefited us a lot.

The microstructures and surface elemental composition of MICP-treated sand samples can be obtained by SEM-EDS analysis. In this study, the microstructures of sand particles with or without MICP treatment were analyzed by SEM, and the image of CaCO3 bridge between sand particles was captured, and then the mechanism of MICP cementation was obtained. For this study, the surface elemental composition of MICP-treated sand samples can be roughly predicted, which may mainly be C, O, Ca, Cl and Si. Besides, the surface element composition of sand samples is not the focus of this paper, so the EDS test was not conducted in this paper.

The paper has been revised according to the specific comments in the manuscript.

Round 2

Reviewer 1 Report

The authors have addressed most of the comments given. However, there are some depictions that should be adjusted. In lines 92 to 95, “As for existing studies on the MICP protection of ….”, the paragraph expresses that large-scale experiments could reflect more real situations than small-scale experiments. However, readers cannot find any discussion about the “difference between large-scale and small-scale experiments” in this study. Thus, it is recommended that the paragraph should be adjusted.

Author Response

Point 1: The authors have addressed most of the comments given. However, there are some depictions that should be adjusted. In lines 92 to 95, “As for existing studies on the MICP protection of ….”, the paragraph expresses that large-scale experiments could reflect more real situations than small-scale experiments. However, readers cannot find any discussion about the “difference between large-scale and small-scale experiments” in this study. Thus, it is recommended that the paragraph should be adjusted.

Response 1: Thank you very much for your comment.

When the slope scale is too small in the model tests, the boundary effect is more obvious than that of the large-scale tests, and a small slope scale may lead to inaccurate sensor measurement in the tests, thus causing large test errors. In addition, a smaller slope scale may result in a steeper slope, which cannot reflect the erosion pattern of gentle slopes such as beaches, and it is difficult to capture the wave change patterns on the slopes.

It has been adjusted in the text.

Reviewer 3 Report

I have read the 2nd version of the manuscript and i believe that it may proceed with publication. 

Author Response

Point 1: I have read the 2nd version of the manuscript and i believe that it may proceed with publication. 

Response 1: Thank you very much for your opportunity to our works.